# Unsupervised Segmentation in NSCLC: How to Map the Output of Unsupervised Segmentation to Meaningful Histological Labels by Linear Combination?

Cleo-Aron Weis [1,*,†] , Kian R. Weihrauch [1,†], Katharina Kriegsmann [1,2,‡] and Mark Kriegsmann [3,‡]

1    Institute of Pathology, University Medical Centre Mannheim, Medical Faculty Mannheim, Heidelberg University, 68167 Mannheim, Germany; kianw@umich.edu (K.R.W.); katharina.kriegsmann@med.uni-heidelberg.de (K.K.)
2    Department of Hematology, Oncology and Rheumatology, University Hospital Heidelberg, 69120 Heidelberg, Germany
3    Institute of Pathology, University Medical Hospital Heidelberg, Heidelberg University, 69120 Heidelberg, Germany; mark.kriegsmann@med.uni-heidelberg.de
*    Correspondence: cleo-aron.weis@medma.uni-heidelberg.de; Tel.: +49-621-383-4072
†    Current address: Institute of Pathology, Medical Faculty Mannheim, Heidelberg University, 68167 Mannheim, Germany.
‡    These authors contributed equally to this work.

**Abstract:** Background: Segmentation is, in many Pathomics projects, an initial step. Usually, in supervised settings, well-annotated and large datasets are required. Regarding the rarity of such datasets, unsupervised learning concepts appear to be a potential solution. Against this background, we tested for a small dataset on lung cancer tissue microarrays (TMA) if a model (i) first can be in a previously published unsupervised setting and (ii) secondly can be modified and retrained to produce meaningful labels, and (iii) we finally compared this approach to standard segmentation models. Methods: (ad i) First, a convolutional neuronal network (CNN) segmentation model is trained in an unsupervised fashion, as recently described by Kanezaki et al. (ad ii) Second, the model is modified by adding a remapping block and is retrained on an annotated dataset in a supervised setting. (ad iii) Third, the segmentation results are compared to standard segmentation models trained on the same dataset. Results: (ad i–ii) By adding an additional mapping-block layer and by retraining, models previously trained in an unsupervised manner can produce meaningful labels. (ad iii) The segmentation quality is inferior to standard segmentation models trained on the same dataset. Conclusions: Unsupervised training in combination with subsequent supervised training offers for histological images here no benefit.

**Keywords:** histopathology; lung cancer; supervised segmentation; unsupervised segmentation





## 1. Introduction

After the emergence of immunohistochemistry in the 1980s, molecular pathology in the 2000s, and next-generation sequencing in the 2010s, the implementation of image analysis tools into the methodical arsenal of pathology appears to be the next level of development. Digital Pathology, Computational Pathology, and Pathomics are several names for this new branch of expertise, and each term represents a slightly different focus [1–3]. Pathomics, for example, focuses on the extraction of image features that can act as biomarkers in the context of, e.g., neoplastic diseases. In this context, image segmentation is one of the early but essential steps. On the basis of segmented images, image features are extracted and used in further analysis [1]. With machine learning-based image segmentation techniques such as convolutional neuronal networks (CNNs), high-quality and reliable image segmentation is possible. These CNN-based segmentation

approaches typically comprise four development phases: Phase 1 is the creation of a well-labelled dataset; phase 2 is the choice of the model architecture; phase 3 is the design or choice of an appropriate loss function; and phase 4 is choosing or defining an appropriate optimiser [4]. For phase 1, in a usual supervised setting, to avoid overfitting, typically, large annotated datasets are necessary. Creating a representative, large training database tends to be tedious, especially the segmentation tasks; therefore, good datasets are scarce [3,5–7]. To overcome this limitation, several publicly available databases are available online—for example, the Atlas of Digital Pathology [8]. Unfortunately, such databases do not help with more specific questions than segmenting different, non-neoplastic tissues. In addition, rare entities cannot be covered. Many technically different methods have been implemented to overcome the dependency on laboriously generated huge databases. These methods either reduce the number of annotated data needed or are completely independent of labelled data. In addition to approaches based on generated features, of particular interest in this study are machine learning methods that learn the features independently [5,9,10]. On one hand, some methods apply machine learning on small datasets, such as few-shot learning [11,12] or zero-shot learning [13,14]. On the other hand, there are completely unsupervised learning methods for classification or segmentation tasks [5].

Against this background, recent publications by Kanezaki et al. on unsupervised image segmentation are of substantial interest. They describe a framework to train CNN segmentation models that is completely unsupervised [15,16].

In this work, (i) we tested this approach for the segmentation of non-small cell lung carcinoma in tissue microarrays as an example. Furthermore, (ii) we addressed the problem that unsupervised segmentation leads to undefined labels. To map the labels by the unsupervised training to known, meaningful labels (e.g., adenocarcinoma), we tested a second training step with a small human-labelled dataset.

## 2. Materials and Methods

### 2.1. Data Collection and Management

Whole-slide tissue specimens of formalin-fixed paraffin-embedded tumour tissue and tissue microarrays (TMAs) were retrieved (Institute of Pathology, Medical Faculty Heidelberg, Heidelberg University) and used in a completely anonymous manner. No patient information—for example, age or sex—was included. Only the histological diagnoses (e.g., normal lung tissue, adenocarcinoma of the lung, squamous cell carcinoma of the lung) were used. This study was approved by the local ethics committee (#S-207/2005 and #S315/2020).

### 2.2. Whole-Slide Image and TMA Image Preparation

The whole tissue sections (haematoxylin–eosin(HE)-stained) and TMAs (HE-stained and stained by immunohistochemistry (IHC) for panCK) were scanned by a Leica whole-slide scanner or by a PreciPoint M8-scanner. The resulting whole-slide images were saved in the .svs format. For model training and validation, the TMA cores were automatically cropped and saved to 2600 × 2600 pixel-sized images by using QuPath implemented functions [17]. The whole-slide images were automatically cropped into tiles of the same size by a QuPath script published by Peter Bankhead.

### 2.3. Training and Validation Dataset (Dataset #1)

Dataset #1 was used for model training and validation. It is based on IHC-HE-TMA core pairs. In this case, every core has a clinical label (normal tissue, adenocarcinoma, squamous cell carcinoma). This dataset was created in a multi-step approach: Step #1: From the included TMA paraffin block, two subsequent sections were produced: the first was HE-stained and the second was IHC-stained (panCK). Both slides were scanned, and the TMA cores were extracted as described above. Step #2: On the basis of their location on the TMA grid, the HE- and the IHC-stained cores can be assigned to each other (e.g., TMA grid position A-1 in HE stain corresponds to TMA grid position A-1 in panCK stain). Next,

these images, containing a single TMA core each, were registered, resulting in the IHC-HE-TMA core pair. For registration, the airlab tool published by Sandkuehler et al. was used (https://github.com/airlab-unibas/airlab accessed on 1 February 2022). Step #3: The IHC-positive area of each image (containing epithelium) was extracted by using a combination of colour deconvolution [18] and thresholding, resulting in a map for background tissue and IHC-positive tissue. Based on the clinical annotation (every TMA grid position is assigned to one case), the IHC-positive areas are assigned to the defined labels: 1 non-tumourous tissue (NT), 2 adenocarcinoma (ADC), and 3 squamous cell carcinoma (SqCC). IHC-negative areas are assigned based on thresholding to the labels background (0) and non-tumourous tissue (1). Notably, based on this assignment approach, IHC-positive epithelium in normal tissue is labelled 1 together with the IHC-negative tissue in the same cases. The label ratio between the area per label is approximately 19.2 (background (BG)) to 4.7 (normal tissue or non-tumourous tissue (NT)) to 1.5 (adenocarcinoma (ADC)) to 1.0 (sqqmous cell carcinoma (SqCC)). Because of the image pair production in steps #1 and #2, the labels produced in step #3 can be used for the HE-stained images. By doing so, the advantage here is that segmentation data are produced without the need for human experts to have laboriously drawn each class per TMA core. However, this advantage is at the cost of errors due to, for example, poor registration or false thresholding between IHC-negative and -positive areas. The multi-stage process only produces a rough visual inspection of the results. In summary, dataset #1 contained n = 247 images (n = 108 for NT, n = 84 for ADC, n = 55 for SqCC). Nine examples are shown in Figure 1. This dataset was used for the training and validation of the modified and retrained unsupervised models (hereafter Kanezaki models) and the supervised model (a UNet-Variant). Dataset #1 is available at HeiData: https://heidata.uni-heidelberg.de/privateurl.xhtml?token=0129f05c-b1a7-4927-a841-2440eb0b3cc4.

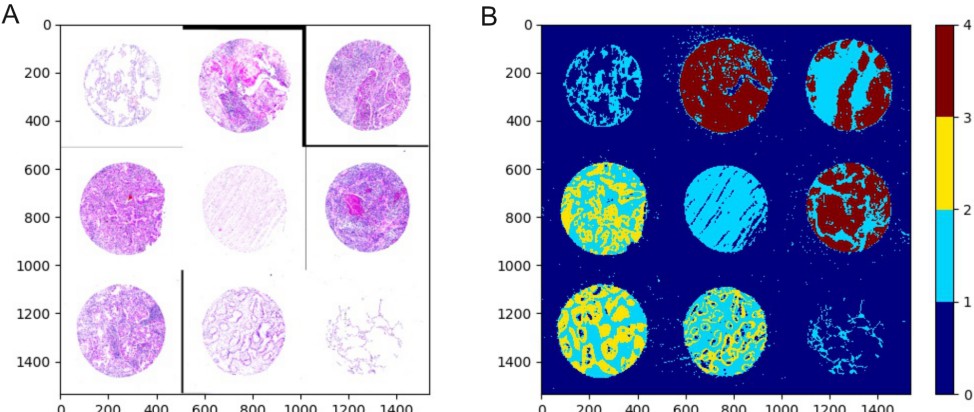

**Figure 1. Example for training and validation dataset (dataset #1).** Based on registered pairs of HE- and IHC-stained sections from TMA cores, tumour segmentation or rather tumour mask generation is performed by a combination of colour deconvolution and thresholding. (**A**) Composite image of nine HE-stained TMA cores. For each of the three classes (NT, ADC, SqCC), there are three images. (**B**) Corresponding IHC(panCK)-stained images were registered on the HE-stained cores. Based on the IHC-positive area and the diagnosis per core, the according image and image regions were labelled: 0 background, 1 normal tissue or non-tumourous (NT), 2 adenocarcinoma (ADC), and 3 squamous cell carcinoma (SqCC)

*2.4. Testing Dataset (Dataset #2)*

Dataset #2 was used for model testing only and was based on manual segmentation (examples shown in Figure 2). Therefore, TMA images were manually annotated, segmented, and further prepared in QuPath [17]. For this manual segmentation, the following labels were defined (in accordance with the definitions for dataset #1): 0 background, 1 non-tumourous tissue (NT), 2 adenocarcinoma (ADC), and 3 squamous cell carcinoma (SqCC). The ratio between the area per label was approximately 3.7 (background) to 1.2 (NT)

to 1.1 (ADC) to 1.0 (SqCC). Dataset #2 contained n = 40 images (n = 3 for NT, n = 18 for ADC, n = 19 for SqCC) and is available at HeiData: https://heidata.uni-heidelberg.de/privateurl.xhtml?token=0129f05c-b1a7-4927-a841-2440eb0b3cc4.

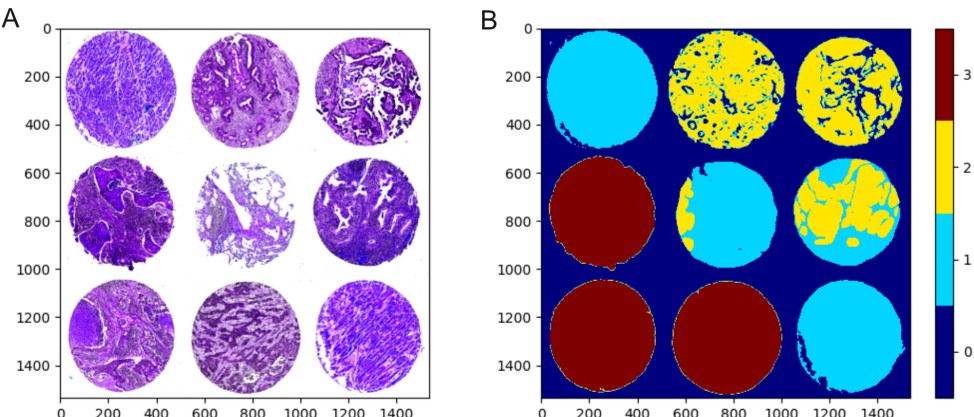

**Figure 2. Example for testing dataset (dataset #2).** Example of human-labelled ground truth images used for testing the models. (**A**) Composite image of nine TMA cores. For each of the three classes (NT, ADC, Sqcc), there were three images. (**B**) Composite image of corresponding labelled images with 0 for background, 1 for normal tissue or non-tumourous tissue (NT), 2 for adenocarcinoma (ADC), and 3 for squamous cell carcinoma (SqCC).

### 2.5. Model Training

Machine learning was performed in Python with PyTorch [19]. For supervised learning, the Segmentation Models toolbox from Yakubovskiy et al. [20] was used (https://github.com/qubvel/segmentation_models.pytorch, accessed on 1 February 2022). For unsupervised training, the models and scripts from Kanezaki et al. [15] were adapted (https://github.com/kanezaki/pytorch-unsupervised-segmentation, accessed on 1 February 2022) and used.

### 2.6. Loss Functions

Against the background of unbalanced labels and heterogeneously shaped objects, different loss functions selected from the plethora of published functions were used. These loss functions are differently well suited for imbalanced datasets. Furthermore, the loss functions are differently well suited for different models [21,22] (Table 1).

**Table 1. List of different loss functions here tested.** A set of loss functions is tested against the background of imbalanced labels and heterogeneous objects.

| Loss Function | Source |
| --- | --- |
| Cross-Entropy Loss | Pytorch implementation [19] |
| Dice Loss | Pytorch implementation [19] |
| Focal Loss | pytorch-toolbelt implementation [23] |
| Tversky Loss | pywick implementation [24] |
| Boundary loss | proposed by Bokhovkin et al. [25] |
| Surface Loss with Cross-Entropy Loss | proposed by Kervadec et al. [26] |
| Surface Loss with Focal Loss | proposed by Kervadec et al. [26] |

### 2.7. Segmentation Quality Assessment

The segmentation quality per image was evaluated by calculating the accuracy and the F1 score, each in its scikit-learn implementation [27]. As ground truth for the calculations, the validation set (see Section 2.3) and the test set (see Section 2.4) were used.

## 3. Results

### 3.1. How Can Labels from a Model Trained in an Unsupervised Manner Be Converted to Meaningful Labels?

For unsupervised image segmentation, Kanezaki et al. (Figure 3) published a training approach in 2018 based on similarity [15] and another in 2020 based on differentiable feature clustering [28]. Here, we use the first approach and the framework described for it, which we refer to as the Kanezaki framework. It starts with a high number of classes and minimises the label classes in every training epoch, until a predefined number of classes is reached. The label classes are merged based on the similarity in the segmented image region. Finally, there are a given number of label classes segmented per image. However, these labels do not correspond to meaningful labels. Which label class (e.g., 1) belongs to which histological structure (e.g., alveolar epithelium) is unclear. We mapped such a model to defined classes by a two-step process.

First, the Kanezaki training approach was reproduced using the histological images available. The excised HE-stained cores were used in random order for training. As we have described, the approach of Kanezaki et al. starts with a predefined number of labels—in our case, 100. Next, at each epoch, the number of labels was reduced or the labels were merged. The training process ended when the previously specified expected number of labels was reached or undercut. Because, in our setting, the classes 'background', 'tumour stroma', 'squamous cell carcinoma', and 'adenocarcinoma' were expected, the number of expected labels was set to 10.

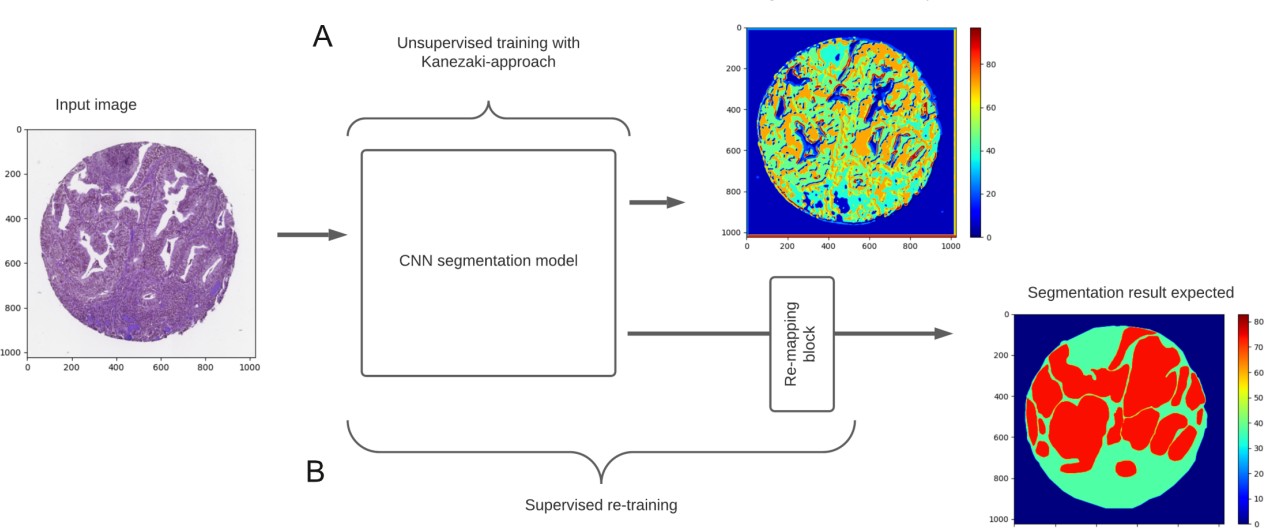

**Figure 3. Schematic of the remapping approach.** Kanezaki et al. described an approach for unsupervised segmentation [15]. This leads to meaningless morphological labelling. For example, the epithelial and stromal structures are segmented. In this manner, homogeneous labels (in terms of texture, for example) are created. Depending on the resolution, this can lead to tumour formation and splitting of the tumour stroma into different partial labels. The main hypothesis of the underlying work is that labels such as 'adenocarcinoma' are composed of a distinct set of morphological labels produced by unsupervised training. The approach described herein is divided into two main parts. (**A**) First, a CNN model (e.g., consisting of several convolutional and batch normalization blocks) is trained in an unsupervised manner, as described by Kanezaki et al. This training was performed on the image batches to ensure that all classes were represented. (**B**) An additional block was added to the model to map the classes of the model to patho-histological labels. The mapping was trained in a supervised manner. There were four components for the label frequency vectors per TMA core.

For the CNN model, a simple model composed of a linear combination of convolution and batch normalisation blocks was used, as described by Kanezaki et al. [15]. This simple

CNN model is henceforth referred to as concise_CNN (visualised in Figure A1A). Other more complex segmentation models, such as UNets or FCNs, have also been trained in this framework. However, only concise models with a few layers (called concise_UNET and concise_FCN; visualised in Figure A1B,C) converged. Standard UNet-variants such as those published in Yakubovskiy et al. (henceforth called Standard_UNET) cannot be trained [20].

Second, to map the labels to meaningful labels, a fully connected layer is added to the model previously trained in an unsupervised setting. This layer is supposed to map the learned labels to given, defined labels, such as 'stroma'. The extended model is then retrained again in a supervised setting on a small, labelled dataset. This two-step approach is performed for three image sizes (256 × 256, 512 × 512, and 1024 × 1024 pixels), to test whether the image or object size affects the segmentation performance.

### 3.2. Do Different Loss Functions Affect Retraining?

The used lung cancer datasets (dataset #1 based on IHC annotations (Section 2.3) and dataset #2 based on manual segmentation (Section 2.4)) are highly heterogeneous. For example, the background area (label 0) was three-times more frequent than the other three labels (1–3) in the overall dataset. In a single image, the ratio of, e.g., adenocarcinoma to stroma can easily exceed 1 to 10. Furthermore, the shape and histological characteristics of tumour formations of one entity (e.g., SqCC) can be diverse. To compensate for the imbalanced dataset with regard to the area per label, we compared different error functions and metrics.

The loss functions are as follows: (1) the PyTorch-implemented weighted cross-entropy loss function [19]; (2) the dice loss and (3) the focal loss function (with the pytorch-toolbelt implementation [23]); (4) the Tversky loss function (with its pywick implementation [24]); (5) the boundary loss function proposed by Bokhovkin et al. [25]; and (6) the surface loss function proposed by Kervadec et al. [26].

As a readout, the segmentation quality was measured by calculating the accuracy and the F1 score. These parameters were assessed for the validation dataset (being 0.25 for dataset #1) and the test dataset (dataset #2). Notably, there is a morphological or quality difference between IHC-based and manual segmentation. Thus, for the models, it is a certain transfer task, because the training, validation, or test data differ.

Independent of the image size, the segmentation quality reaches its highest value when the cross-entropy loss function is used alone or in combination (for both validation and testing). Notably, the segmentation quality was only moderate even for the best models. For example, for the validation dataset (see Table 2), we observed unbalanced cross-entropy with an accuracy of $0.88 \pm 0.11$ and an F1 score of $0.63 \pm 0.19$; for balanced cross-entropy, we observed an accuracy of $0.82 \pm 0.13$ and an F1 score of $0.47 \pm 0.16$; and for surface loss (in combination with balanced cross-entropy), as described by Kervadec et al. [26], we observed an accuracy of $0.86 \pm 0.22$ and an F1 score of $0.39 \pm 0.12$).

### 3.3. In Comparison, What Are the Results of an Often-Used Segmentation Model Trained in a Supervised Manner?

For comparing the segmentation quality, a standard UNet-variant [20] was trained. This model was trained and validated on dataset #1 (Section 2.3) and tested on dataset #2 (Section 2.4) under the same conditions as described above for the Kanezaki models. Different image sizes and error functions were used to visualise their effects on the segmentation quality (measured with accuracy and the F1 score).

**Table 2. Retrained unsupervised segmentation results.** An adapted Kanezaki model was retrained on different image sizes (256 × 256 and 512 × 512 pixels) with five different loss functions: (1) cross-entropy, (2) dice loss, (3) focal loss, (4) Tversky loss, and (5) boundary loss function. The validation set corresponds to 0.25 from dataset #1 (compare Section 2.3) and the test set corresponds to the entire dataset #2 (compare Section 2.4). As a metric for the segmentation quality, the accuracy and F1 score are calculated.

| Image Size | Loss Function | Validation Set (Accuracy/F1) | | Test Set (Accuracy/F1) | |
|---|---|---|---|---|---|
| 256 | Cross-Entropy (balanced) | $0.82 \pm 0.13$ | $0.47 \pm 0.16$ | $0.63 \pm 0.21$ | $0.37 \pm 0.09$ |
| | Cross-Entropy (unbalanced) | $0.88 \pm 0.11$ | $0.63 \pm 0.19$ | $0.59 \pm 0.21$ | $0.43 \pm 0.16$ |
| | Surface Loss (with Cross-Entropy) | $0.86 \pm 0.22$ | $0.39 \pm 0.12$ | $0.51 \pm 0.17$ | $0.33 \pm 0.10$ |
| | Surface Loss (with Dice Loss) | $0.32 \pm 0.12$ | $0.28 \pm 0.11$ | $0.44 \pm 0.23$ | $0.30 \pm 0.11$ |
| | Focal Loss | $0.88 \pm 0.12$ | $0.63 \pm 0.21$ | $0.59 \pm 0.20$ | $0.43 \pm 0.18$ |
| | Tversky Loss | $0.87 \pm 0.12$ | $0.62 \pm 0.21$ | $0.57 \pm 0.19$ | $0.39 \pm 0.16$ |
| | Dice Loss | $0.86 \pm 0.12$ | $0.60 \pm 0.21$ | $0.61 \pm 0.16$ | $0.37 \pm 0.21$ |
| | Boundary Loss | $0.13 \pm 0.07$ | $0.15 \pm 0.06$ | $0.27 \pm 0.17$ | $0.17 \pm 0.06$ |
| 512 | Cross-Entropy (balanced) | $0.86 \pm 0.12$ | $0.53 \pm 0.17$ | $0.60 \pm 0.11$ | $0.40 \pm 0.08$ |
| | Cross-Entropy (unbalanced) | $0.87 \pm 0.12$ | $0.56 \pm 0.18$ | $0.60 \pm 0.16$ | $0.37 \pm 0.10$ |
| | Surface Loss (with Cross-Entropy) | $0.86 \pm 0.11$ | $0.56 \pm 0.20$ | $0.66 \pm 0.16$ | $0.41 \pm 0.10$ |
| | Surface Loss (with Dice Loss) | $0.28 \pm 0.09$ | $0.26 \pm 0.08$ | $0.33 \pm 0.10$ | $0.25 \pm 0.07$ |
| | Focal Loss | $0.88 \pm 0.12$ | $0.58 \pm 0.21$ | $0.64 \pm 0.18$ | $0.36 \pm 0.11$ |
| | Tversky Loss | $0.88 \pm 0.11$ | $0.56 \pm 0.20$ | $0.60 \pm 0.17$ | $0.36 \pm 0.10$ |
| | Dice Loss | $0.88 \pm 0.11$ | $0.54 \pm 0.20$ | $0.60 \pm 0.15$ | $0.37 \pm 0.10$ |
| | Boundary Loss | $0.13 \pm 0.08$ | $0.14 \pm 0.06$ | $0.20 \pm 0.11$ | $0.15 \pm 0.07$ |

*3.4. How Does the Training Dataset Size Affect the Segmentation Quality of the Models Trained under Unsupervised and Supervised Conditions?*

To test whether the segmentation performance of the models tested depends on the size of the training data as expected, the modified concise_CNN and the UNet-model were pretrained with eight subsets of different size from the previous datasets (see x-axis in Figure 4: The first subset, named selection, was a manual image selection with three images per diagnosis (NT, ADC, and SqCC) from dataset #2 (see Section 2.4). The other eight subsets were named with 1.0, 0.75, 0.5, 0.25, 0.1, 0.05, and 0.01, respectively, after the fractions from dataset #1 (see Section 2.3). In addition, to test the effect of the pretraininig of the UNet-model, two UNet-models differing in the means of pretraining were tested: one naive for histological images and henceforth referred to as UNet_naive (orange bars in Figure 4), and the other with a ResNet-model [20] pretrained on a image tile classification task with the classes normal tissue, adenocarcinoma, and squamous cell carcinoma, hereafter referred to as UNet_histo (green bars in Figure 4).

All models were trained with the surface loss function (with the combination of boundary loss and cross-entropy loss) [26] for 50 epochs. All image tiles used for training and testing had a size of 256 × 256 pixels.

Segmentation quality was assessed based on the testing images (dataset #2), as in the prior section.

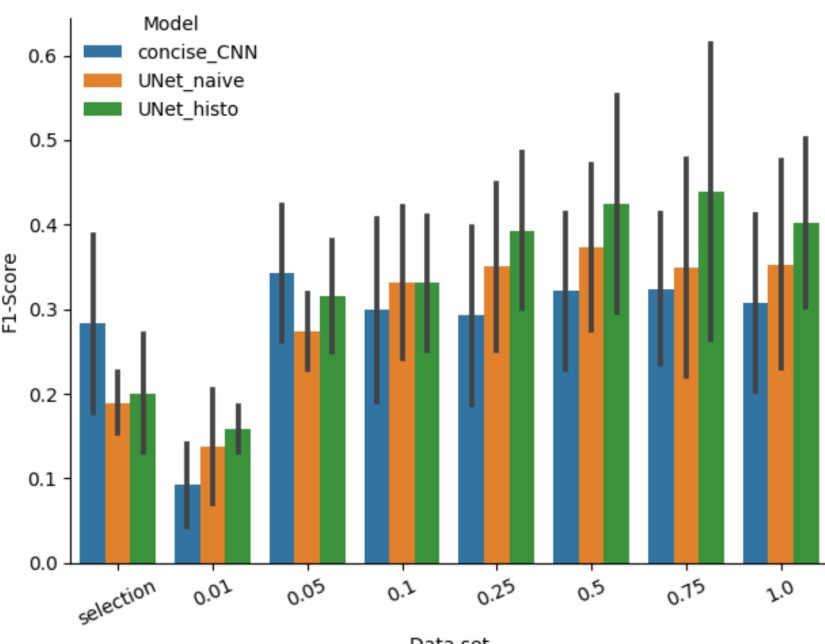

**Figure 4. Effect of the number of images used for training.** Three models were trained on eight different datasets. The three models are: a small CNN variant as described by Kanezaki et al. [15] (called concise_CNN), trained in an unsupervised approach and then modified and retrained as described in this work to produce histologically meaningful labels; two UNet models [20] with a ResNet backbone pretrained on a tissue classification task (called UNet_histo) and with a ResNet backbone without pretraining (called UNet_histo). The eight datasets are a manual selection (with nine images, three per diagnosis; compare Figure 1) from dataset #1 (called selection), and seven increasingly larger fractions from dataset #2 (called 0.01, 0.05, 0.1, 0.25, 0.5, 0.75, and 1.0) Subsequently, the different models were tested using dataset #2. As a segmentation metric, the F1 score is plotted.

The models trained on dataset #2 had equal segmentation quality, as in the prior section. For the Concise_CNN, the accuracy was $0.49 \pm 0.22$ and the F1 score was $0.31 \pm 0.10$. For the UNet_naive, the accuracy and the F1 score were $0.53 \pm 0.20$ and $0.35 \pm 0.12$. Finally, for the UNet_histo, the accuracy was $0.63 \pm 0.18$ and F1 score was $0.40 \pm 0.10$ (right end of the box plot in Figure 4). Notably, no statistically significant differences were observed in the accuracy and F1 score for the models in the range 0.1 to 1.0: for 0.1 of dataset #1, the F1 score was $0.30 \pm 0.11$; for the UNet_naive, it was $0.33 \pm 0.09$; and for the UNet_histo, it was $0.31 \pm 0.07$. Only for the small fractions (0.01 and the nine images from the selection) were the models' performance reduced (left end of the box plot in Figure 4). For the selection with nine images, there was a trend for better performance of the Kanezaki model, since its accuracy ($0.43 \pm 0.23$) and F1 score ($0.29 \pm 0.10$) were not reduced as much as those for the UNet models (for UNet_naive, $0.26 \pm 0.06$ and $0.20 \pm 0.04$, respectively, and for UNet_histo, $0.13 \pm 0.19$ and $0.20 \pm 0.07$, respectively)

*3.5. Does the Model Architecture Trained in an Unsupervised Manner Influence the Segmentation Quality?*

Kanezaki et al. [15] demonstrated that with their approach, different CNN models can be trained. As we have described, complex models such as the UNet variant by Yakubovskiy et al. [20] do not converge. However, simple model variants for FCN and UNet can be trained and do converge. To compare three different architectures, we first used the unsupervised Kanezaki approach to train the aforementioned models, which consist of a linear combination of convolution blocks (called concise_CNN; sketched in Appendix A Figure A1A), a relatively simple FCN variant (called concise_FCN; Appendix A Figure A1B), and a relatively simple UNet variant (called concise_UNet; Appendix A Figure A1C).

Next, these three models were retrained in a supervised setting (as described in Section 4.1). To test if there is an advantage of such pretrained models for smaller datasets, we performed retraining by using two training datasets: (i) a manual selection of nine images (three per NT, ADC, and SqCC; called selection) and (ii) the entire dataset #1.

Regarding segmentation quality measurement, these models were again tested on dataset #2 (the testing dataset).

For very small retraining datasets (selection; n = 9 images), the Concise_CNN shows better results if only the last layers are retrained (accuracy 0.53 ± 0.21 and F1 score 0.34 ± 0.11). For the more complex concise_FCN (accuracy 0.50 ± 0.24 and F1 score 0.30 ± 0.10) and concise_UNet (accuracy 0.43 ± 0.11 and F1 score 0.24 ± 0.09), the models only show moderate segmentation quality, if the entire models are retrained (Figure A2A).

For larger retraining datasets (dataset #1; n = 247 images (with 0.8 for training and 0.2 for validation)), there was no significant difference for all three models (Figure A2B). The best model was the simple_FCN. The accuracy was 0.63 ± 0.21 and the F1 score was 0.38 ± 0.10 when only the last layers were retrained, and the accuracy was 0.61 ± 0.15 and the F1 score was 0.387 ± 0.07 when all layers were retrained. The worst model was the simple_CNN. For retraining only the last layers, the accuracy was 0.57 ± 0.17 and the F1 score was 0.36 ± 0.10. For retraining all layers, the accuracy was 0.57 ± 0.19 and the F1 score was 0.37 ± 0.10.

Notably, segmentation results were more than 0.1 worse than the results for the complex UNet model variants, such as the Standard_UNET (compare Table 3) by Yakubovskiy et al. [20].

**Table 3. Supervised segmentation results.** A UNet model was trained on different image sizes (256x256 and 512x512 pixels) with five different loss functions: (1) cross-entropy, (2) dice loss, (3) focal loss, (4) Tversky loss, and (5) boundary loss function. The validation set corresponds to 0.25 from dataset #1 (compare Section 2.3) and the test set corresponds to the entire dataset #2 (compare Section 2.4). As a metric for the segmentation quality, the accuracy and the F1 score are calculated.

| Image Size | Loss Function | Validation Set (Accuracy/F1) | | Test Set (Accuracy/F1) | |
|---|---|---|---|---|---|
| 256 | Cross-Entropy (balanced) | 0.86 ± 0.13 | 0.52 ± 0.20 | 0.71 ± 0.17 | 0.43 ± 0.08 |
| | Cross-Entropy (unbalanced) | 0.89 ± 0.11 | 0.64 ± 0.23 | 0.64 ± 0.19 | 0.45 ± 0.16 |
| | Surface Loss (with Cross-Entropy) | 0.87 ± 0.12 | 0.57 ± 0.17 | 0.86 ± 0.17 | 0.41 ± 0.08 |
| | Surface Loss (with Dice Loss) | 0.78 ± 0.18 | 0.41 ± 0.08 | 0.57 ± 0.16 | 0.34 ± 0.07 |
| | Focal Loss | 0.88 ± 0.11 | 0.63 ± 0.23 | 0.70 ± 0.17 | 0.42 ± 0.08 |
| | Tversky Loss | 0.88 ± 0.12 | 0.64 ± 0.23 | 0.70 ± 0.19 | 0.42 ± 0.09 |
| | Dice Loss | 0.88 ± 0.12 | 0.54 ± 0.65 | 0.71 ± 0.22 | 0.51 ± 0.21 |
| | Boundary Loss | 0.13 ± 0.14 | 0.06 ± 0.16 | 0.24 ± 0.04 | 0.19 ± 0.03 |
| 512 | Cross-Entropy (balanced) | 0.86 ± 0.12 | 0.47 ± 0.09 | 0.70 ± 0.14 | 0.44 ± 0.08 |
| | Cross-Entropy (unbalanced) | 0.90 ± 0.10 | 0.66 ± 0.22 | 0.66 ± 0.15 | 0.42 ± 0.10 |
| | Surface Loss (with Cross-Entropy) | 0.90 ± 0.12 | 0.59 ± 0.19 | 0.68 ± 0.12 | 0.42 ± 0.08 |
| | Surface Loss (with Dice Loss) | 0.90 ± 0.09 | 0.54 ± 0.14 | 0.66 ± 0.14 | 0.40 ± 0.17 |
| | Focal Loss | 0.91 ± 0.10 | 0.61 ± 0.19 | 0.66 ± 0.15 | 0.40 ± 0.08 |
| | Tversky Loss | 0.90 ± 0.11 | 0.65 ± 0.21 | 0.66 ± 0.19 | 0.43 ± 0.13 |
| | Dice Loss | 0.90 ± 0.10 | 0.64 ± 0.21 | 0.64 ± 0.16 | 0.39 ± 0.08 |
| | Boundary Loss | 0.11 ± 0.06 | 0.12 ± 0.05 | 0.23 ± 0.11 | 0.15 ± 0.05 |

### 3.6. Are the Labels Learned in an Unsupervised Fashion Already Meaningful?

As we have described, the models trained without supervision produce a predefined number of labels that are not directly connected to labels defined by humans. These labels are based on texture or morphological similarity. To test whether these labels alone correlate to the known structures (e.g., tumour glands) or more to the diagnoses of non-tumourous tissue (NT), adenocarcinoma (ADC), or squamous cell carcinoma (SqCC), in this context, the frequency of these labels per diagnosis was examined.

The basic idea was that each TMA core has its own label composition or frequency that correlates with the diagnosis. For example, a TMA core from a case with ADC should contain (only) the labels NT and/or ADC. To test this assumption in principle, for human-generated labels, we plotted the label composition (frequency of labels per TMA core) against the known diagnosis per TMA kernel (see A1-2 in Figure 5). Plotting the label frequency per diagnosis (A1 in Figure 5) or running a principal component analysis (PCA) to compare the frequency vectors per TMA core (A2 in Figure 5) verifies this assumption. As expected, cases with, for example, a diagnosis of ADC differ in that only in these cases does the label ADC occur alongside the image background (BG) and non-tumourous tissue (NT).

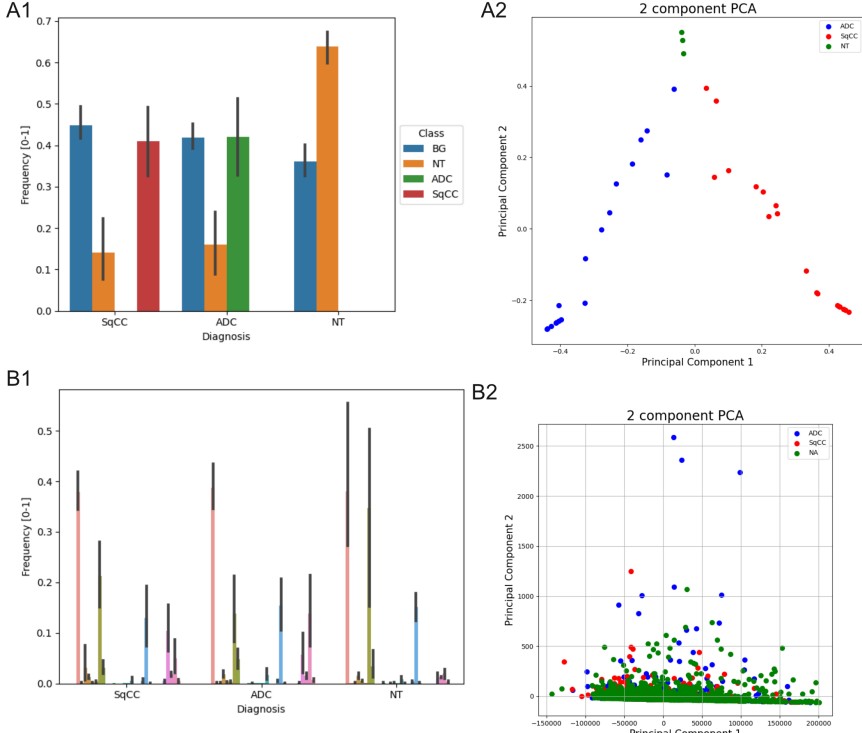

**Figure 5. Segmentation results for different models trained.** Kanezaki et al. [15] described a method for training CNN models in an unsupervised fashion, resulting in some labels that are not assigned to certain structures by a human. For testing whether these labels or a combination are already meaningful, the label frequency per TMA core was analysed with respect to the known diagnosis per core (normal tissue (NT), adenocarcinoma (ADC), and squamous cell carcinoma (SqCC)). (**A1,A2**) For testing whether the label frequency per TMA core could correlate with the diagnosis, the label frequency of the manually annotated TMA cores was examined. (**A1**) shows the label frequency (background (BG), normal tissue (NT), adenocarcinoma (ADC), and squamous cell carcinoma (SqCC)) plotted against the diagnosis. (**A2**) shows a PCA (two components) for the label frequency vector per TMA core. (**B1,B2**) For the labels produced by a simple CNN model (previously called concise_CNN) trained in an unsupervised manner, the label frequency was also analysed in regard to the known diagnosis per TMA core. (**B1**) shows the frequency distribution per label and diagnosis. (**B2**) shows a PCA (two components) for the label frequency vectors per TMA core.

In the next step, we examined the label frequency for the labels generated by the model after unsupervised training. Here, it can be seen that neither plotting the label frequency per diagnosis (B1 in Figure 5)) nor a PCA analysis (B2 in Figure 5) show a reliable correlation between label composition and diagnosis. A distinction between ADC and SqCC is not possible on this basis. Based on these plots, only TMA cores with and without tumour infiltration can be distinguished. A comparable analysis for the standard UNet [20] trained in a supervised setting also showed no sharp separation of diagnosis based on label frequency (compare Appendix A Figure A3). This fits well with the overall moderate segmentation quality of the models trained and validated. In the unsupervised trained or retrained models, the small network size might be causative. In the standard UNet, the training database may not be sufficiently large.

## 4. Discussion

Digital and complex medical data are available from various medical specialities. For patients with tumours, for example, there are molecular data, and radiological and pathological image data [10,29]. The analysis of these vast data from one field—or, better, combined—leads to an opportunity to find next-generation, data-driven biomarkers [29]. In pathology, Pathomics is the subdiscipline dedicated to mapping image data to clinical information such as nodal status. In other words, new image-based biomarkers are being sought in Pathomics. In this context, the segmentation of histological images (e.g., in tumour and stroma) is an early and major step in many projects [1,4]. For supervised segmentation approaches, the scarcity of large, properly annotated datasets is a common obstacle [3,7,30]. Not only is annotation tedious per se, but in many cases, the number of available images is limited. For example, managing thymoma, a rare disease, significantly limits the number of cases to be included [31]. Data augmentation techniques alone have usually not solved the problem of small numbers [7,10]. In addition to supervised training approaches, there are weakly supervised and unsupervised approaches that can help to overcome this constraint. However, mapping the results produced by unsupervised approaches to meaningful labels is a non-trivial task. Against this background, we tested, in the complex setting of the distinction between pulmonary (solid-growing) ADC and (non-keratinising) SqCC [32], whether a segmentation model could (i) be trained in an unsupervised approach and (ii) modified and retrained in a supervised setting to produce meaningful histological labels such as 'ADC' or 'SqCC'. In the best case, these labels should handle the aforementioned non-trivial distinction between solid-growing ADC and non-keratinising SqCC. (iii) We compared the the segmentation results to standard segmentation models for the same datasets.

(Ad i), we show that unsupervised image segmentation techniques or training frameworks as described by Kanezaki et al. can be used for the unsupervised segmentation of histological images (Figure 3A) [15,16]. Here, only aspects such as the ratio of filter size to object size in the image need to be considered.

(Ad ii), we demonstrate that these models can be extended by another block, which can, after a second supervised training, remap (by a linear combination) the produced labels to meaningful labels such as 'ADC' or 'SqCC' (Figure 3B). Our new contribution is that a simple linear combination of the different labels previously recognised based on unsupervised training is applied to predict difficult labels such as 'ADC'.

(Ad iii), finally, we compare the results to conventional training approaches and demonstrate that our approach of remapping the labels is not superior to conventional supervised learning. It is indeed inferior and there are only limited settings where it can be useful.

### 4.1. Unsupervised Segmentation in Pathology and the Problem of Obtaining Meaningful Labels (Ad i)

Image segmentation is, for many projects in the realm of Digital Pathology or Pathomics, an important early step. There is a legion of different approaches that, based on

the training setting, can be broadly categorised as fully supervised, weakly supervised, and unsupervised approaches. For the supervised training approaches, the necessary annotations are time-consuming and tedious to produce. Indeed, the shortage of such annotated datasets is a well-known obstacle [3,7,30]. Needing less annotated data for weakly supervised or no annotated data at all for unsupervised methods sounds, in this context, very promising. In addition, by using unsupervised approaches, the need to tailor to every project a well-annotated training set for machine learning models will be reduced to gathering a fitting image collection [10].

Regarding weakly supervised approaches in pathology, generative adversarial networks can be used (after training in a weakly supervised setting) to generate synthetic data based on a small dataset [5,33]. However, this would mean adding a training cost-intensive step before the actual segmentation model training.

Regarding unsupervised image segmentation, in pathology, there are several published approaches [5,9]. These approaches cover a vast methodological spectrum with, for example, the combination of feature extraction and subsequent clustering [9,34] or the application of auto-encoders for classification or staining adaption [5,35–37].

In a nutshell, there are various working, published, easily adaptable, unsupervised approaches for segmenting (histological) images into different morphological regions. This would then overcome the problem of data scarcity. Unfortunately, this advantage brings a new problem. The labels generated based on morphological aspects (e.g., 'blue granular area') cannot in every case be simply mapped to (histologically) meaningful labels such as 'carcinoma'. One solution to this is to assign names to the labels by human experts. For example, we could allow the expert to define blue areas with many small cells as lymphoid infiltration. However, this expert approach only works if exactly one label is generated per annotation. A multiphase process such as a tumour consisting of tumour cells, stroma, and inflammatory infiltrate, etc., will not be nameable in this manner. For such multiphase entities, the true annotation can be considers as a combination of the morphological labels. However, this linear combination is too simplistic for many areas, as the context is then missing. For example, a homogeneous, blue area can be part of the sky or a blue car. In this regard, there are works that use graphs to include the neighbourhood relationships of the individual labels. For example, Pourian et al. used graphs of regions to combine the visual and spatial characteristics of different image parts to meaningful image-part groupings [38]. Alternatively, Wigness et al. used local graphs to combine labels in image regions [39].

Our approach, by contrast, is a simple linear combination of the different labels generated by unsupervised learning based on morphological similarity (by a adding a fully connected layer to a CNN model; see B in Figure 3). This linear combination is in analogy to the pathological thinking of tissue or organs as a combination of different structures such as epithelium, stroma, blood vessels, etc. [40,41]. However, this approach ignores neighbourhood relations or local aspects.

*4.2. CNN Models Previously Trained in an Unsupervised Manner Can Be Adapted to Produce Meaningful Histological Labels (Ad ii)*

As we have discussed, unsupervised training approaches can be used for histological images, but they produce distinct regions or labels based on morphology (e.g., reddish area with little texture) without histologically meaningful labels (such as, e.g., 'fibrosis'). We have successfully trained several CNN model variations (a combination of convolutional blocks (called concise_CNN), a shallow UNET variant (called concise_UNET), and a shallow FCN variant (called concise_FCN)) in an unsupervised approach, as described by Kanezaki et al. [15,16]. The produced image regions, or rather labels, however, are not mapped to the conventional histological structures. For instance, gland structures are composed of an epithelial layer (one label) and the luminal space (another label). Pathologists would usually annotate these structures together as a gland, in analogy to the typical thinking of tissues and organs as combinations of a limited number of substrata [40,41].

For remapping the labels produced by the model trained in an unsupervised manner, we added a block (a fully connected layer) and then trained it on mapping the labels to human-produced annotations (compare B in Figure 3). However, this again necessitates the presence of (a small amount of) annotated data. Notably, this approach is therefore no longer an unsupervised but a weakly supervised approach. By adding another block and retraining, we can show that a model can produce meaningful annotations. However, compared to other segmentation models (such as the UNet implementation by Yakubovskiy et al. [20]) trained on the same dataset, the approach proposed here leads to inferior results.

### 4.3. The Combination of Unsupervised and Subsequent Supervised Label Mapping Is Not Better than Conventional CNN-Based Segmentation (Ad iii)

The inferior segmentation results in combination with again the need of a labelled dataset argue against the herein proposed remapping of labels by adding an additional block and by retraining with a small annotated dataset (compare Figure 3).

There are several potential explanations for the rather moderate segmentation results of the proposed remapping approach:

(1) The model complexity is maybe not fitting with the task. Of note, the used framework for unsupervised training described by Kanezaki et al. [15,16] only works with shallow CNN models. Large models such as the UNet model implemented by Yakubovskiy et al. [20] do not converge.

(2) Another idea would be that the ratio of the CNN filter size to the object size in the images is either too small or too large. Therefore, we tested different images sizes ($256 \times 256$, $512 \times 512$, and $1024 \times 1024$ pixels) and found no significant difference. Likewise, it would be possible in principle that the CNN models trained in this way were too shallow. However, this is contradicted by the fact that the tumour sub-type differentiation also did not work well in the UNet models used by other groups [20], which are frequently used and perform well.

(3) The task itself is non-trivial since neither models adapted as described nor standard segmentation models can make the distinction between different tumour types, particularly between (non-keratinising) SqCc and (solid) ADC. In the used dataset, on which a work on tumour classification has been published recently [32], the models can only distinguish background from tissue and normal tissue from tumour parts. Of course, the models trained supervised perform better (see Tables 2 and 3); however, their segmentation results are also only moderate with assigning mixed labels (such as ADC and SqCC) per tumour infiltration. This could be due to several reasons. For example, the task of distinguishing between a non-glandular growing ADC and a non-keratinising SqCC is non-trivial, even for an experienced pathologist, on the basis of HE-stained images alone. Moreover, unlike the previously published work on this dataset for classification [32], where each image must be assigned to one class, now, each pixel must be assigned the correct label.

(4) Finally, maybe the labels or morphological clusters segmented by the models after unsupervised training are found within non-neoplastic and neoplastic structures. For example, glandular structures are found in both. To test this, we looked at the distribution of morphological labels generated by such a model compared to the diagnoses per TMA core (see Figure 5). Based on plotting for every TMA core the frequency of the labels background (BG), normal tissue (NT), adenocarcinoma (ADC), and squamous cell carcinoma (SqCC) in a two-dimensional PCA, we were able to classify the TMA cores into the three classes of normal tissue, adenocarcinoma, and squamous cell carcinoma (see Figure 5A1,A2). Interestingly, for the models initially trained unsupervised and then retrained and for the UNets trained supervised, based on the label distribution, such a classification is not possible (see Figure 5B1,B2 and Appendix A Figure A3A,B). This is an argument for the assumption that solely morphological labels are not enough for the herein analysed task, in line with the discussion in the section above.

The limitations described above raise the question of whether the method can be improved. Increasing the complexity of the models pretrained in an unsupervised manner alone seems not promising, since even complex models such as UNets are not able to adequately solve the task. The most promising approaches, in view of the work of Pourian et al. [38] or Wigness et al. [39], seem to involve the neighbourhood when combining the individual labels into meaningful histological annotations. This should certainly be followed up in future work. Moreover, the approach described here should certainly be tested on a simple histological task as a proof of principle.

### 4.4. Are There Arguments for Using the Herein Proposed Remapping Approach?

Regarding the only moderate segmentation quality on one hand and the greater effort of retraining on the other hand, one could ask if there are arguments for using such an approach. Having a well-annotated dataset at hand, there are no arguments against using a standard supervised training setting. Of course, large and good datasets are rather scarce for histology [3,5–7]. The approach proposed herein, which, in combination, is more akin to weakly supervised approaches, might provide an initial advantage if there is only a dataset of limited size. Moreover, in a scenario in which there is only a small dataset, one could also consider using methods such as generative adversarial networks to produce synthetic data, on which then the segmentation model is trained [5,33].

**Author Contributions:** Conceptualisation, C.-A.W., K.K. and M.K.; methodology and formal analysis, C.-A.W.; data generation and curation, K.R.W., C.-A.W., K.K. and M.K.; manuscript writing and preparation, C.-A.W., K.K. and M.K. All authors have read and agreed to the published version of the manuscript.

**Institutional Review Board Statement:** The study was approved by the ethics committee of the Medical Faculty Heidelberg, Heidelberg University (#S-207/2005 and #S315/2020).

**Data Availability Statement:** Image data (for training, validation, and testing) along with ground truth are available at HeiData: https://heidata.uni-heidelberg.de/privateurl.xhtml?token=0129f05c-b1a7-4927-a841-2440eb0b3cc4. The code for the work is available at GitHub: GitHub-Link/The repo is created when the publication title is clear so that it can be generated under the same name.

**Acknowledgments:** The authors gratefully acknowledge the data storage service SDS@hd supported by the Ministry of Science, Research and the Arts Baden-Württemberg (MWK) and the German Research Foundation (DFG) through grant INST 35/1314-1 FUGG and INST 35/1503-1 FUGG. Furthermore, the authors also thank the IT department staff of the Medical Faculty Mannheim and especially Bohne-Lang for supervising the computer administration and infrastructure.

**Conflicts of Interest:** The authors declare no conflicts of interest.

### Abbreviations

The following abbreviations are used in this manuscript:

| | |
|---|---|
| MDPI | Multidisciplinary Digital Publishing Institute |
| CNN | Convolutional Neuronal Network |
| PCA | Principal Component Analysis |
| GAN | Generative Adversial Network |

## Appendix A

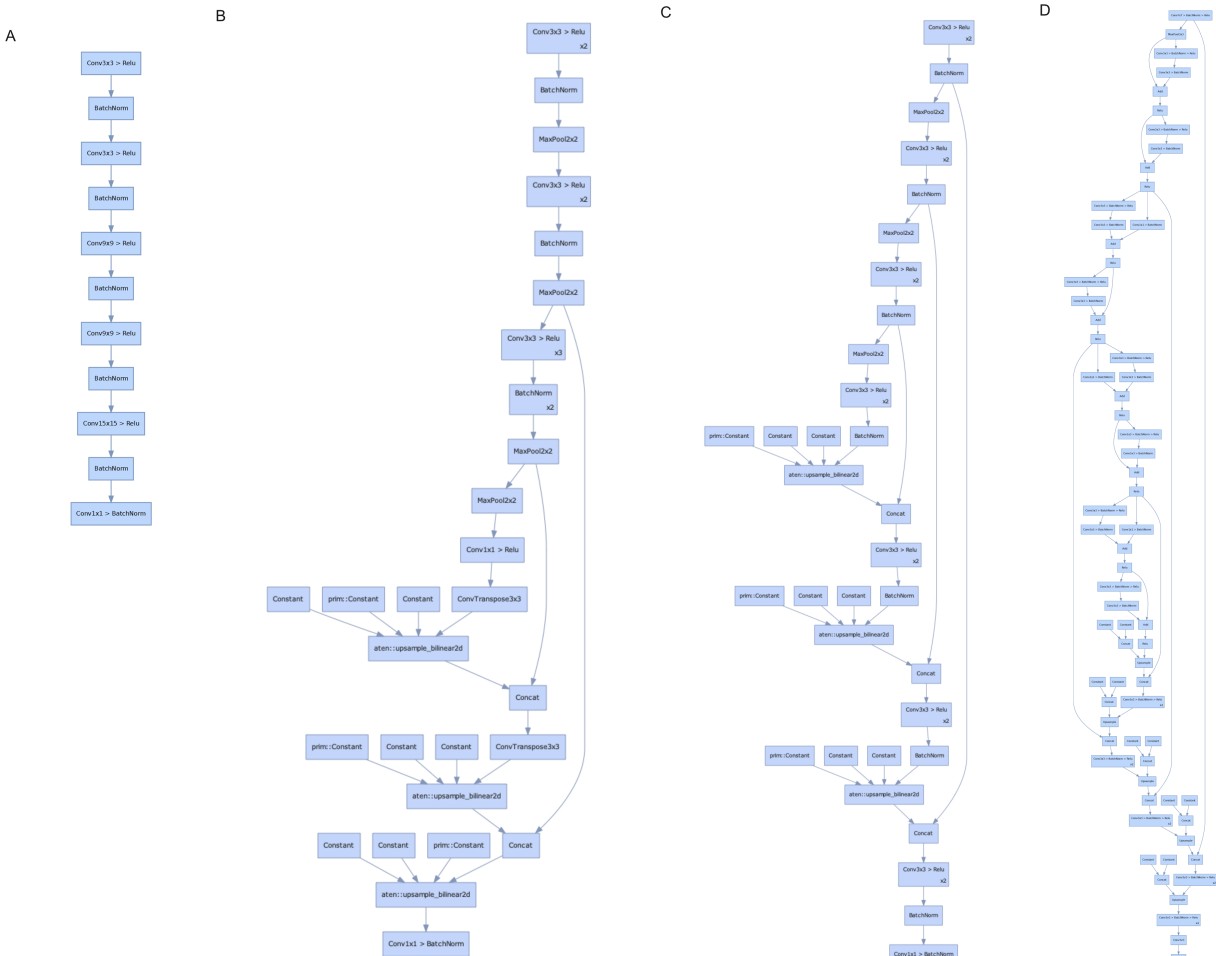

**Figure A1. Schematic representation of the models used.** The models (**A–C**) were first trained in an unsupervised fashion and subsequently trained on mapping the labels to meaningful labels. In contrast, the model in (**D**) was trained solely in a supervised fashion. (**A**) Concise_CNN: Schematic plot of the simple CNN model used by Kanezaki et al. [15], which is composed of a linear combination of convolutional, ReLu, and batch normalization layers. (**B**,**C**) Concise_UNET and Concise_FCN: Schematic representation of concise UNET and FCN model variants. (**D**) Standard_UNet: Schematic representation of the UNet model designed by Yakubovskiy et al. [20], here used for supervised segmentation.

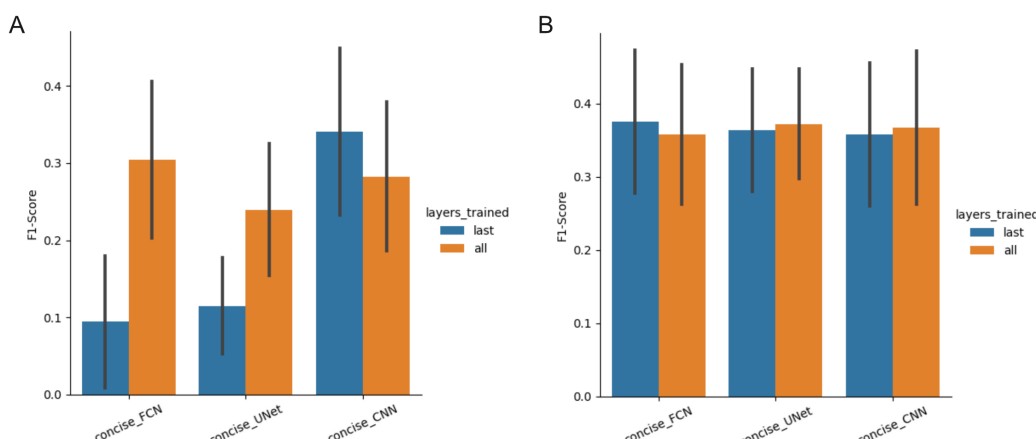

**Figure A2. Segmentation results for different models trained.** Kanezaki et al. [15] described a method for training CNN models in an unsupervised fashion. In this approach, different CNN models can be trained. Here, a simple CNN model composed of several convolution blocks (simple_CNN), a simple FCN variant (simple_FCN), and a simple UNet variant (simple_UNet) are trained and retrained on two datasets (**A**,**B**). Furthermore, in the retraining, only the last layers (last) or the entire model (all) are retrained. (**A**) The models are retrained on a selection of nine images (three images per diagnosis: NT, ADC, and SqCC). (**B**) The models are retrained on the entire dataset #2.

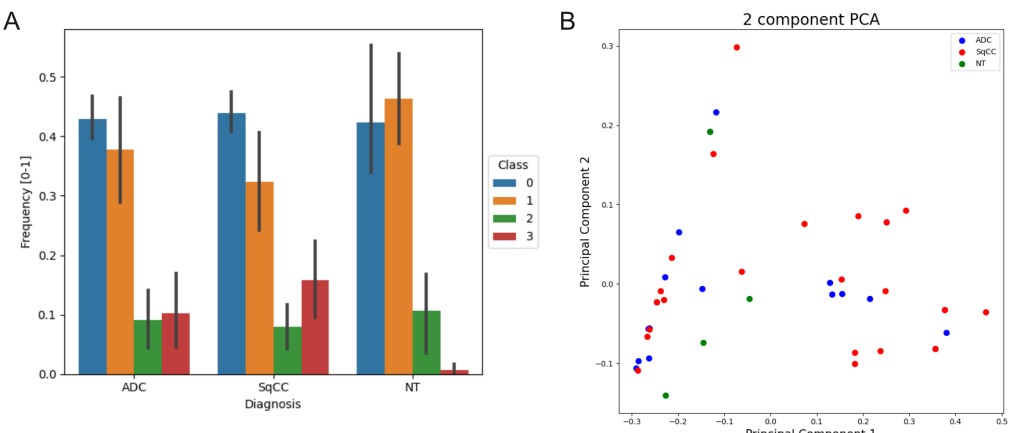

**Figure A3. Segmentation results for a UNet model.** The labels produced per image by a UNet model [20] trained in a supervised fashion are plotted. (**A**) shows the label frequency (background (BG), normal tissue (NT), adenocarcinoma (ADC), and squamous cell carcinoma (SqCC)) plotted against the diagnosis. (**B**) shows a PCA (two components) for the label frequency vector per TMA core.

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
