# Peer review of "Unsupervised Segmentation in NSCLC: How to Map the Output of Unsupervised Segmentation to Meaningful Histological Labels by Linear Combination?"

_applsci, doi:10.3390/app12083718_

Round 1
Reviewer 1 Report
In this manuscript, the authors utilized convolutional neural networks for image segmentation. Furthermore, the authors have presented a comparison of supervised and unsupervised approaches to deal with the problem of data insufficiency.
The proposal of this work presents several deficiencies that need to be addressed.
The paper needs thorough proofreading.
Lines 3 and 266-267, We have “Its quality relies on the quality and quantity of …”. And “The segmentation quality usually correlates to the quality and quantity of the training data.” This is incorrect. The authors should check these statements.
Justify the novelty of this paper.
The authors must provide related work in a separate section.
Present architecture diagram of the proposed method.
Please put the citation/source of the dataset so that the reader could easily find it.
Page 3 line 96, Please provide a proper reference of the dataset.
Page 9 lines 186-192, The paragraph is unreadable. Please revise and explain it properly.
Figure 4 horizontal label is missing. The authors should provide a proper figure and recheck the description of the figure.
Page 10 lines 212-214, The authors didn’t provide the relevant figures.
Page 11 lines 259 and 260, We have “A comparable analysis for the standard-UNET (not shown) also showed no sharp separation”. Check it.
Page 11 line 287, There is an unnecessary symbol of ‘?’.
A detailed performance evaluation must be presented. The authors should compare their proposed method results with the most recently published works to show the novelty of this paper.
The authors should significantly improve the presentation of their work.
The authors should improve their English.
Author Response
Dear reviewer,
Thank you very much for your remarks, comments etc. We have gladly taken them up. Accordingly, we have rewritten the manuscript and adapted the work.
Please find below your itemized comments with our respective answers.
Remarks Reviewer # 1
In this manuscript, the authors utilized convolutional neural networks for image segmentation. Furthermore, the authors have presented a comparison of supervised and unsupervised approaches to deal with the problem of data insufficiency.
The proposal of this work presents several deficiencies that need to be addressed.
Point #1.1: The paper needs thorough proofreading.
Thank you for this advice. The paper was sent to a proof-reading service.
Point #1.2: Lines 3 and 266-267, We have “Its quality relies on the quality and quantity of …”. And “The segmentation quality usually correlates to the quality and quantity of the training data.” This is incorrect. The authors should check these statements.
Both sentences are rephrased to highlight that for a proper generalization typically large datasets are needed.
Point #1.3: Justify the novelty of this paper.
Thank you very much for this advice. In our paper we tested, if the meaningful label produced by a model trained without supervision, can be mapped to pathohistological labels. We proposed to add a layer for re-mapping. This approach is much simpler than ones proposed by other groups.
Point #1.4: The authors must provide related work in a separate section.
This comment is related to point 1.3. We added in the discussion a section on related works (Can unsupervised approaches produce meaningful labels?) to provide a background against which our work can be seen.
Point #1.5: Present architecture diagram of the proposed method.
Thank you for this advice. We shifted the original figure 1 (showing the models used) to the supplement and added a new figure illustrating our approach.
Point #1.6: Please put the citation/source of the dataset so that the reader could easily find it.
Page 3 line 96, Please provide a proper reference of the dataset.
We have now included the proper reviewer-links. These links are private and only supposed for the review process. If the paper is accepted, these links need to be changed.
Point #1.7: Page 9 lines 186-192, The paragraph is unreadable. Please revise and explain it properly.
Thank you for mentioning this section. It was not easy to understand. We have modified the section and the figure in this regard.
Point #1.8: Figure 4 horizontal label is missing. The authors should provide a proper figure and
recheck the description of the figure.
Thank you for mentioning that. We have adapted the figure for the missing label. Furthermore, we have re-phrased the figure description in the context of the above-mentioned issue #1.7
Point #1.9: Page 10 lines 212-214, The authors didn’t provide the relevant figures.
Thank you. There was a typo in the reference link. We have corrected it. The model figure is moved to the supplement.
Point #1.10: Page 11 lines 259 and 260, We have “A comparable analysis for the standard-UNET (not shown) also showed no sharp separation”. Check it.
The according figure (bar plot and PCA) for a UNet trained in a supervised setting is added to the supplement (Fig. A3).
Point #1.11: Page 11 line 287, There is an unnecessary symbol of ‘?’.
Thank you. We have removed it.
Point #1.12: A detailed performance evaluation must be presented. The authors should compare their proposed method results with the most recently published works to show the novelty of this paper.
Thank you for mentioning this. We added a paragraph to the method sections, where we describe the metrics used.
Regarding comparison to other, published approaches, in the manuscript we use a UNet-implementation published by Yanokowsiky et al. (Yakubovskiy, P. Segmentation Models Pytorch. Publication Title: GitHub repository.) as reference model. This model is trained in an usual supervised setting on the same datasets.
Regarding showing and discussing the novelty of using a linear combination of morphological labels to produce meaningful labels (in histology), we re-wrote the entire discussion.
Point #1.13: The authors should significantly improve the presentation of their work.
Thank you. We have modified the discussion extensively to focus on the manuscript on the novelty. Furthermore, we have added a new figure and a new supplemental figure. The remaining figures and their descriptions have been also modified. And we shifted several figures to the supplement to make the main body more concise.
Pont #1.14: The authors should improve their English.
Thank you. We have submitted the manuscript to a language editing service; and adapted it accordingly.
Reviewer 2 Report
Automatic segmentation is a promising image analysis tools in the area of pathology and others. In this manuscript, the authors didn’t propose an effective convolutional neuronal network (CNN) approach for image segmentation on Lung carcinoma slides. Besides, an unsupervised segmentation approach proposed by Kanezaki was adopted for quantitative comparison with other supervised approaches. As mentioned in Lines 275-277, “Third, we compare the results to conventional training approaches and demonstrate that the herein presented approach of re-mapping the labels is not superior to conventional supervised learning” and in Lines 320-321, “Compared to other segmentation models trained on the same data set, the approach proposed here leads to inferior results.” The novelty and impact of this work is very limited. The flow of the manuscript is difficult to follow. Neither the model of the CNN approach nor the trend of the comparison is explicitly depicted. Therefore, this manuscript is not appropriate to be published.
Author Response
Dear reviewer,
Thank you very much for your remarks, comments etc. We have gladly taken them up. Accordingly, we have rewritten the manuscript and adapted the work.
Please find below your itemized comments with our respective answers.
Reviewer / Point #2: Automatic segmentation is a promising image analysis tools in the area of pathology and others. In this manuscript, the authors didn’t propose an effective convolutional neuronal network (CNN) approach for image segmentation on Lung carcinoma slides. Besides, an unsupervised segmentation approach proposed by Kanezaki was adopted for quantitative comparison with other supervised approaches. As mentioned in Lines 275-277, “Third, we compare the results to conventional training approaches and demonstrate that the herein presented approach of re-mapping the labels is not superior to conventional supervised learning” and in Lines 320-321, “Compared to other segmentation models trained on the same data set, the approach proposed here leads to inferior results.” Point #2.1 The novelty and impact of this work is very limited. Point #2.2 The flow of the manuscript is difficult to follow. Point #2.3 Neither the model of the CNN approach nor the trend of the comparison is explicitly depicted. Therefore, this manuscript is not appropriate to be published.
Thank you very much for your comments especially regarding to the classification of the results. We have modified the manuscript in this regard. Especially in the introduction and the discussion, we have better embedded our contribution in the overall context (ad point #2.1 and #2.2. Furthermore, we have emphasized our (for histological images) new approach (ad point #2.1). Regarding the model and approach (ad point #2.3) we have added a new workflow sketch.
To our knowledge, mapping the labels produced by a model trained in an unsupervised way to meaningful labels by linear combination is new (ad point #2.1). Albeit it sounds promising, we can show for our data set, that it does not solve the task. We tried to emphasize this negative, but still interesting result within the entire manuscript body.
Reviewer 3 Report
The manuscript is very poorly edited. Numerous editing mistakes can be found throughout the manuscript. The motivations/needs for the work were not sufficiently presented, and it was not clear what is the new progress in the manuscript.
Author Response
Dear reviewer,
Thank you very much for your remarks, comments etc. We have gladly taken them up. Accordingly, we have rewritten the manuscript and adapted the work.
Please find below your itemized comments with our respective answers.
Reviewer / Point #3:
Point #3.1: The manuscript is very poorly edited. Numerous editing mistakes can be found throughout the manuscript.
Thank you. We have re-edited the manuscript and submitted it to a professional proofreading service. Please note the numerous changes highlighted in colour (blue text or yellow background colour) throughout the entire manuscript.
Point #3.2: The motivations/needs for the work were not sufficiently presented, and it was not clear what is the new progress in the manuscript.
Thank you for this valuable assessment. It showed us that we need to significantly adjust or change the focus of the introduction and especially the discussion. Now it leads to main issue of “how to map labels produced from a model trained without supervision to meaningful histological labels”.
Round 2
Reviewer 1 Report
The authors have addressed all of my comments and I recommend accepting the paper.
Author Response
Thank you again for your input on our work. It has definitely benefited (regarding display of results, language, discussion etc.) from them.
Reviewer 3 Report
While the manuscript has been significantly improved, some editing error remains. Please fix the question marks in table 1.
Author Response
Thank you for your comments and thoughts on our manuscript. These really helped improve it. The question marks in the table are removed. There was a formatting issue.